

# Key regulators of vascular calcification in chronic kidney disease: Hyperphosphatemia, BMP2, and RUNX2

Xinhua Liang[1], Yankun Li[1], Peng Wang[2] and Huafeng Liu[2]

[1] Affiliated Hospital of Guangdong Medical University, Guangdong Provincial Key Laboratory of Autophagy and Major Chronic Non-communicable Diseases, Key Laboratory of Prevention and Management of Chronic Kidney Disease of Zhanjiang City, Institute of Nephrology, Zhanjiang, Guangdong Province, China
[2] Affiliated Hospital of Guangdong Medical University, Guangdong Provincial Key Laboratory of Autophagy and Major Chronic Non-communicable Diseases, Zhanjiang, Guangdong, China

## ABSTRACT

Vascular calcification is quite common in patients with end-stage chronic kidney disease and is a major trigger for cardiovascular complications in these patients. These complications significantly impact the survival rate and long-term prognosis of individuals with chronic kidney disease. Numerous studies have demonstrated that the development of vascular calcification involves various pathophysiological mechanisms, with the osteogenic transdifferentiation of vascular smooth muscle cells (VSMCs) being of utmost importance. High phosphate levels, bone morphogenetic protein 2 (BMP2), and runt-related transcription factor 2 (RUNX2) play crucial roles in the osteogenic transdifferentiation process of VSMCs. This article primarily reviews the molecular mechanisms by which high phosphate, BMP2, and RUNX2 regulate vascular calcification secondary to chronic kidney disease, and discusses the complex interactions among these factors and their impact on the progression of vascular calcification. The insights provided here aim to offer new perspectives for future research on the phenotypic switching and osteogenic transdifferentiation of VSMCs, as well as to aid in optimizing clinical treatment strategies for this condition, bearing significant clinical and scientific implications.

Corresponding authors
Peng Wang, wangpeng@gdmu.edu.cn
Huafeng Liu, liuhf@gdmu.edu.cn

# INTRODUCTION

Currently, cardiovascular complications are the leading cause of death among patients with chronic kidney disease (CKD) and those with end-stage renal disease (ESRD), accounting for 30%–50% of all fatalities (*Vogt, Haffner & Leifheit-Nestler, 2019*; *Yamada & Giachelli, 2017*). In the later stages of CKD, the incidence of cardiovascular complications significantly increases due to the decline in renal function. This decline impairs the kidney's ability to manage the metabolism of minerals such as calcium and phosphorus in the body, thereby accelerating vascular calcification (VC) and arteriosclerosis (*Hénaut et al., 2018*; *Liu et al., 2014*; *Shroff, Long & Shanahan, 2013*). VC in CKD patients can occur in both the intima and the media of the vascular walls, and these two types of calcifications may

occur concurrently, leading to an increased cardiovascular mortality rate in CKD patients (*Anonymous, 2009*; *Viegas et al., 2019*). There is ample clinical evidence to suggest that VC is significantly associated with the incidence and mortality of cardiovascular complications in patients with CKD and ESRD. It is considered one of the most effective independent predictors of cardiovascular risk in CKD patients (*Alani, Tamimi & Tamimi, 2014*; *Blacher et al., 2001*; *Cozzolino et al., 2018*). According to statistics from the United States Renal Data System (USRDS), the risk of cardiovascular complication-related mortality in ESRD patients undergoing dialysis is five to 30 times higher compared to the healthy population. Furthermore, the process of vascular calcification in CKD patients begins several years earlier (*Collins et al., 2015*; *Goodman et al., 2000*).

It is evident that vascular calcification poses a significant threat to the survival rate and quality of life of patients with chronic kidney disease. However, there is currently no effective clinical treatment available to reverse the progression of vascular calcification in patients with chronic kidney disease. Therefore, this review will provide a detailed introduction to the regulatory mechanisms and signaling molecules that have a significant impact on the progression of vascular calcification in chronic kidney disease. This will offer potential new strategies for subsequent downstream mechanism research and exploration of clinical treatment strategies.

## SURVEY METHODOLOGY

We have conducted an extensive search of the relevant literature published in last 20 years using various scientific search engines such as PubMed, Google Scholar, and Baidu Scholar, with search terms "chronic kidney disease with vascular calcification" and "osteogenic transdifferentiation", to analyze the role of related osteogenic transcription factors such as BMP2 and RUNX2 in mediating the osteogenic transdifferentiation of vascular smooth muscle cells in the progression of chronic kidney disease with vascular calcification. During the retrieval process, we focused on the regulatory mechanisms of hyperphosphatemia, BMP2 and RUNX2 in chronic kidney disease with vascular calcification and the interconnection between the three. After carefully screening the retrieved literature, we only included publications that are relevant to the subject. Studies that did not meet the current inclusion criteria, were irrelevant to the topic, conference records, editorials, and reviews with insufficient data were excluded from the inclusion criteria.

## VASCULAR CALCIFICATION DURING THE COURSE OF CHRONIC KIDNEY DISEASE

VC, or vascular calcification, refers to the abnormal deposition of inorganic salts composed of calcium and phosphate ions in cardiovascular tissues such as the walls of blood vessels, the heart, and heart valves. It is commonly seen in aging, diabetes and end-stage renal disease. As the renal function of patients with chronic kidney disease declines, the disorder in phosphorus excretion leads to bone metabolism disorders such as secondary hyperparathyroidism. The reduction in renal phosphorus excretion

and increased secretion of calcium and phosphorus metabolism-regulating hormones such as parathyroid hormone (PTH) and FGF23 (Fibroblast Growth Factor 23) jointly promote disorders in the metabolism of minerals like calcium and phosphorus in the body, leading to renal osteodystrophy. Vascular calcification caused by calcium and phosphorus deposition within the vascular walls is one of the manifestations of renal bone metabolism disorders. The most crucial role in the pathological process leading to VC formation is played by the osteogenic transdifferentiation of VSMCs. This process involves the transformation of contractile type vascular smooth muscle cells into secretory osteoblast-like cells, destruction of intracellular actin filaments, and deposition of large amounts of minerals such as calcium and phosphorus in the muscular layer, ultimately leading to adverse outcomes such as vascular wall stiffness and luminal narrowing (*Moe et al., 2003*; *Smith, 2016*). The prevalent clinical view is that VC is one of the characteristics of advanced chronic kidney disease-mineral and bone disorder (CKD-MBD). In addition to VC, symptoms of abnormal serum biochemical indicators such as hyperphosphatemia, hypercalcemia, and bone disorders are also present. Among these characteristics, VC is considered a hallmark feature of CKD-MBD (*Moe et al., 2006*; *Yamada & Giachelli, 2017*). Clinically, VC is defined as the pathological deposition of calcium phosphate crystals in the cardiovascular system. In late-stage CKD patients, there are several circulating factors that induce VC, collectively referred to as the uremic milieu (*Demer & Tintut, 2008*; *Hénaut et al., 2018*). It has been found that due to the decline in renal excretory function, the patient's serum contains very high levels of uremic toxins and inorganic phosphates, both of which have a strong inductive effect on VC. In addition, there are multiple circulating pro-VC factors, which ultimately induce the deposition of inorganic salts of calcium and phosphate in the intimal and medial layers of blood vessels (*Ciceri & Cozzolino, 2021*). The impact of VC on cardiovascular outcomes in CKD patients is highly related to the location of the mineral deposits. Intimal calcification reflects, to some extent, the burden of arteriosclerotic plaques and is a significant predictor of cardiovascular event mortality in late-stage CKD patients (*Viegas et al., 2019*). Intimal calcification can ultimately lead to vascular wall stiffening, increased pulse pressure, left ventricular hypertrophy, and even heart failure, adversely affecting the quality of life and cardiovascular outcomes in the CKD population (*Raggi et al., 2007*). Therefore, a deeper understanding of the mechanisms of pathological calcification during the late stages of CKD is particularly important, in order to reduce the risk of adverse cardiovascular outcomes, improve the quality of life, and extend the life expectancy of the CKD population.

## THE MOLECULAR MECHANISM OF VASCULAR CALCIFICATION

The mechanisms that promote the development of vascular calcification in chronic kidney disease have been found to share similarities with normal physiological bone formation (*Alves et al., 2014*; *Doherty & Detrano, 1994*). These include the transdifferentiation of osteochondrogenic cells, apoptosis of vascular smooth muscle cells, extracellular matrix remodeling, and the release of extracellular vesicles, with the osteochondrogenic

transdifferentiation of VSMCs playing a particularly crucial role (*Lanzer et al., 2014*; *Shanahan et al., 2011*). It has been observed that these various molecular mechanisms can act concurrently in the process of VC, without necessarily being mutually exclusive(*Johnson, Leopold & Loscalzo, 2006*).

Disturbances in the homeostasis of mineral metabolism and high concentrations of phosphate are considered major determinants in the progression of VC in CKD patients (*Schlieper et al., 2016*; *Shanahan et al., 2011*). Phosphate complexes first activate intracellular signaling pathways that precede calcification (*Alesutan et al., 2017*; *Voelkl et al., 2018b*). High serum levels of phosphate upregulate the expression of type III sodium-dependent phosphate transport protein (Pit-1) on VSMC membranes, increasing intracellular inorganic phosphate (Pi) levels. This, in turn, activates intracellular PI3K/AKT and ERK1/2 signaling pathways, inducing an increase in RUNX2 expression and promoting the osteogenic transdifferentiation of VSMCs (*Chavkin et al., 2015*; *Lee, Lee & Jeon, 2020*). Osteogenic transdifferentiation refers to the process by which VSMCs transform into cells with an osteoblast-like phenotype through selective gene expression. During this process, the expression of VSMC-specific proteins such as $\alpha$-smooth muscle actin ($\alpha$-SMA) and smooth muscle protein 22$\alpha$ (SM22$\alpha$) decreases, while the expression of osteoblast/chondrocyte-specific proteins like osteopontin (OPN) and osteocalcin (OCN) increases (*Steitz et al., 2001*).

BMP2 and RUNX2 play extremely important roles in this process. BMP2 is a member of the transforming growth factor (TGF-$\beta$) protein family and has been shown to activate Runx2 in various cell types, playing an essential role in bone formation and repair (*Durham et al., 2018*; *Orth-Alampour et al., 2021*; *Tsuji et al., 2006*). The pivotal role of these pathways highlights potential targets for therapeutic interventions aimed at mitigating VC in patients with CKD.

## THE MAIN CAUSE OF VASCULAR CALCIFICATION: HIGH PHOSPHORUS

Among the various factors that can induce vascular calcification in patients with chronic kidney disease, hyperphosphatemia shows the most significant correlation with VC secondary to CKD, and it is a quintessential feature of the disorder in mineral and bone metabolism associated with CKD (*Block et al., 2004*; *Taniguchi et al., 2013*). Under normal circumstances, the level of serum inorganic phosphorus is mainly regulated by physiological processes such as the absorption of phosphorus by the intestines, filtration and reabsorption by the renal tubules, and the exchange of substances inside and outside cells, among which the role of the kidneys is crucial. With the decline in renal function in the later stages of patients with chronic kidney disease, the kidney's ability to excrete phosphorus decreases, leading to a large accumulation of phosphates in the internal environment. This activates the osteogenic transdifferentiation process within vascular smooth muscle cells, prompting the transformation of vascular smooth muscle cells into an osteoblast-like cell phenotype, increasing intracellular calcium ion levels, and consequently leading to vascular wall calcification, vessel narrowing, and increased vascular stiffness, among other

adverse outcomes (*Tonelli, Pannu & Manns, 2010*). As renal function declines in the later stages of CKD, serum inorganic phosphate levels rise, which then induces a phenotypic transformation of vascular smooth muscle cells into osteoblast-like cells, playing a crucial role in promoting VC (*Luong et al., 2018*; *Voelkl et al., 2018b*). Osteoblast-like cells express a variety of osteogenic transcription factors, such as msh homeobox 2 (MSX2), RUNX2, and Osterix (Osx), with RUNX2 playing a decisive role in VC secondary to CKD (*Alesutan et al., 2017*; *Chen & Moe, 2015*; *Voelkl et al., 2013*). Research by *Speer et al. (2010)* and *Sun et al. (2012)* has found that knocking down the expression of RUNX2 in VSMCs with siRNA can inhibit osteochondrogenic transdifferentiation of VSMCs and VC. Various osteogenic transcription factors can further stimulate VSMCs to express osteoblast-specific proteins, such as osteocalcin, bone morphogenetic protein 2 and alkaline phosphatase (ALP), especially BMP2, which significantly promotes the uptake of phosphate by VSMCs and their phenotypic changes (*Leibrock et al., 2016*; *Li, Yang & Giachelli, 2008*; *Shanahan et al., 1999*).

Furthermore, in the process of VC secondary to CKD, high levels of Pi activate various intracellular signaling pathways that regulate VC. In VSMCs, the uptake of Pi is primarily mediated by type III sodium-dependent phosphate co-transporters Pit1 and Pit2 (*Lang et al., 2013*; *Shanahan et al., 2011*). Activation of the ERK1/2 signaling pathway may be related to downstream signaling of Pit1 and upregulates the expression of RUNX2 and ALP in VSMCs (*Chavkin et al., 2015*; *Li, Yang & Giachelli, 2006*; *Speer et al., 2009*). Recent studies have found that Pit2 may upregulate the expression of osteoprotectin, inhibiting the progression of VC (*Makarović et al., 2015*; *Yamada et al., 2018*). Elevated serum Pi levels also activate the nuclear transcription factor NF-$\kappa$B signaling pathway (*Yoshida et al., 2017*; *Zhao et al., 2012*). Research by *Voelkl et al. (2018a)* has found that the NF-$\kappa$B signaling pathway can promote osteogenic transdifferentiation of VSMCs by upregulating the expression of osteogenic transcription factors MSX2 and RUNX2, thereby inducing an increase in ALP expression. The WNT/$\beta$-catenin signaling pathway is considered pivotal in VC signaling, and its activation upon elevated extracellular Pi levels can upregulate the expression of RUNX2 and Pit1, thereby promoting osteogenic transdifferentiation of VSMCs (*Cai et al., 2016*; *Yao et al., 2015*). Research by *Deng et al. (2016)* and *Martínez-Moreno et al. (2012)* has shown that using specific inhibitors of the Wnt/$\beta$-catenin signaling pathway, such as Dickkopf-related protein 1 (DKK1) and secreted frizzled-related proteins (SFRPs), can inhibit the calcification of VSMCs (*Deng et al., 2016*; *Martínez-Moreno et al., 2012*).

Studies also suggest the involvement of inflammatory cytokines in the regulation of VC secondary to CKD in the context of hyperphosphatemia. Research by *Yamada et al. (2014)* has found that a high phosphate load can induce local inflammation in cultured VSMCs, producing pro-inflammatory cytokines such as tumor necrosis factor $\alpha$, interleukin-1$\beta$(IL-1$\beta$), interleukin-6(IL-6), BMP2 and others (*Agharazii et al., 2015*; *Yamada et al., 2014*). Research by *Lee et al. (2010)* has found that TNF-$\alpha$ can promote osteogenic transdifferentiation of VSMCs by upregulating the expression of MSX2 through the NF-$\kappa$B signaling pathway. Other studies have found that IL-6 can induce osteogenic transdifferentiation of VSMCs through the activation of the WNT/$\beta$-catenin and STAT3

signaling pathways (*Abedin et al., 2006*; *Lin et al., 2016*; *Sun et al., 2017*). Additionally, research indicates that cell apoptosis signaling pathways are activated during the regulation of VC secondary to CKD by hyperphosphatemia. Research by *Qiu et al., (2017)*, *Son et al. (2006)* and *Son et al. (2007)* and others has found that high extracellular Pi levels can downregulate the expression of growth arrest-specific gene 6 (Gas6) and its receptor tyrosine kinase Axl in VSMCs, inhibit the activation of the Gas6/Axl/Akt anti-apoptotic signaling pathway, induce the deactivation of the anti-apoptotic protein Bcl2 and activation of the pro-apoptotic protein Bad, leading to apoptosis in VSMCs (*Qiu et al., 2017*; *Son et al., 2006*; *Son et al., 2007*). AMPK is considered an important upstream regulatory factor for Gas6 expression (*Ma et al., 2019*). Research by *Xu et al. (2017)* has shown that activation of AMP-activated protein kinase (AMPK) can inhibit the calcification of VSMCs, and the activity of AMPK decreases when extracellular Pi levels are elevated.

Therefore, hyperphosphatemia can be said to be a decisive factor in inducing VC secondary to CKD. Further exploration of the mechanism by which hyperphosphatemia regulates VC holds positive significance for the future discovery of diagnostic methods and treatment plans for VC secondary to CKD.

## THE ROLE OF BMP2 IN THE PROCESS OF VASCULAR CALCIFICATION

Bone morphogenetic proteins are members of the transforming growth factor-$\beta$ family, and current research progress indicates that multiple BMPs are involved in the regulatory process of VSMC calcification. For example, BMP2 and BMP4 are considered to be markers and driving factors of vascular calcification, yet the presence of BMP7 seems to help inhibit the progression of VC in the later stages of CKD (*Yang et al., 2020*). Wang EA, Sampath TK, and others were among the first to discover that BMPs promote ectopic bone formation, and later studies found a substantial expression of these proteins at sites of vascular calcification (*Boström et al., 1993*; *Sampath & Reddi, 1981*; *Wang et al., 1990*). Research by *Scimeca et al. (2019)* analyzed calcified plaques from 52 patients with carotid calcification, finding a significant correlation between the expression of BMP2 and the presence of unstable plaques. Other studies proved that BMP2 can enhance osteogenic transdifferentiation of VSMCs through its pro-inflammatory and pro-atherosclerosis effects, thereby promoting VC (*Li, Yang & Giachelli, 2008*; *Yung et al., 2015*). Studies by *Liberman et al. (2011)* showed that BMP2 upregulates the expression of the decisive osteogenic transcription factor RUNX2, promoting pathological calcium and phosphate accumulation in cultured human VSMCs through oxidative stress and endoplasmic reticulum stress mechanisms. *Dalfino et al. (2010)* and *Rong et al. (2014)* proposed that in late-stage CKD patients, the serum levels of BMP2 increase, and extracellular high Pi along with BMP2 can upregulate the expression of osteogenic transcription factors MSX2 and RUNX2 in VSMCs, promoting their transdifferentiation into osteoblast-like cells. This process found that $\beta$-catenin protein links high Pi, BMP2, and the WNT/$\beta$-catenin signaling pathway in the regulation of VC as a whole (*Dalfino et al., 2010*; *Rong et al., 2014*).

In addition, recent years' research has found that BMP2 undergoes epigenetic modifications in the process of regulating VC. It is commonly believed that osteogenic reprogramming refers to the process in which the phenotype of VSMCs is transformed into that of osteoblasts through epigenetic modifications such as DNA methylation, without changing the gene sequence. Research by *Chen et al. (2016)* and *Pons et al. (2009)* indicates that DNA methylation and histone modifications are important regulatory factors in the osteogenic reprogramming process. *Ouyang et al. (2021)* found that, in the VSMCs of the aorta in a CKD mouse model, the expression of the DNA demethylase ALKBH1 was increased. ALKBH1 enhances the binding of the octamer-binding transcription factor Oct4 to the BMP2 promoter through demethylation modification of $N^6$-methyladenine on the BMP2 promoter, thereby upregulating the expression of BMP2 and promoting osteogenic reprogramming of VSMCs as well as VC (*Ouyang et al., 2021*). *Zeng et al. (2021)* found that by inhibiting the activation of the ERK1/2 signaling pathway and upregulating microRNA-126-3p expression in VSMCs cultured from a mouse model of VC, microRNA-126-3p's interference with BMP2 gene expression at the post-transcriptional level was enhanced, thus inhibiting osteogenic transdifferentiation of VSMCs. Moreover, *Pei et al. (2020)* found that microRNA-204 seems to also interfere with the expression of BMP2 genes at the post-transcriptional level during the osteogenic reprogramming of VSMCs. Research by *Yanagawa et al. (2012)* confirmed that microRNA-141 can inhibit BMP2-mediated signal transduction in VSMCs, thereby partially reversing VSMC calcification and osteogenic transdifferentiation.

Therefore, it is evident that BMP2 plays an extremely important role in the regulatory process of VC secondary to CKD. Clarifying the factors affecting the expression of BMP2 will promote the development of clinical treatment methods for this disease.

## THE ROLE OF RUNX2 IN THE PROCESS OF VASCULAR CALCIFICATION

In the process of vascular calcification, Runt-related transcription factor 2 is commonly regarded as a decisive osteogenic transcription factor. Its upregulated expression in cardiovascular tissues is an important regulatory factor for adverse cellular events in the cardiovascular outcomes of patients with chronic kidney disease (*Chen, Zhao & Wu, 2021*). Research by *Engelse et al. (2001)* and *Tyson et al. (2003)* found a positive correlation between arterial calcification scores and upregulated expression of RUNX2 in both human and mouse models of VC. Recent studies have found that RUNX2 seems to act as a downstream molecule of various pro-osteogenic signaling pathways during the progression of VC. Under high phosphate conditions, the expression of RUNX2 in isolated mouse VSMCs is upregulated due to the activation of the PI3K/AKT signaling pathway, and the expression of RUNX2 is reduced when the PI3K/AKT pathway is inhibited using AKT inhibitors (*Byon et al., 2008*; *Deng et al., 2015*). In the research by *Yang et al. (2018)* experiments in both *in vivo* and *in vitro* mouse VSMCs suggested that the activation of the p38-MAPK signaling pathway appears to induce the binding and phosphorylation of RUNX2 with p38 MAPK, thereby enhancing the transcriptional activity of RUNX2 and

promoting VSMC calcification and osteogenic transdifferentiation. *Li et al. (2022)* found that melatonin could inhibit the calcification of VSMCs by suppressing the activation of the NF-$\kappa$B/CREB signaling axis, and thus preventing the binding of NF-$\kappa$B to the RUNX2 promoter region.

Furthermore, in the regulation of VC, RUNX2 undergoes various post-translational modifications that can affect its regulatory functions (*Chen, Zhao & Wu, 2021*). Studies have found that the activated ERK/MAPK and p38-MAPK signaling pathways can mediate the phosphorylation of RUNX2, thereby enhancing its transcriptional activity (*Greenblatt et al., 2010*; *Xiao et al., 2000*; *Yang et al., 2018*). Similarly, the transcriptional coactivator p300, which possesses histone acetyltransferase (HAT) activity, enhances the stability of RUNX2 through acetylation, making it less susceptible to degradation by the ubiquitin-proteasome system and promoting the osteogenic transdifferentiation process in VSMCs (*Jeon et al., 2006*). In addition to the mentioned PTMs of RUNX2, ubiquitination and O-GlcNAcylation (O-linked $\beta$-N-acetylglucosamine) modifications also play important roles in regulating the stability and transcriptional activity of RUNX2, and consequently, regulate osteogenic reprogramming and VC in VSMCs (*Ong, Han & Yang, 2018*; *Wells, Vosseller & Hart, 2001*; *Yamashita et al., 2005*; *Zhao et al., 2004*). Some recent studies have found that microRNAs also regulate the expression of RUNX2 during the osteogenic transdifferentiation process of VSMCs. MicroRNA-32 has been shown to suppress the expression of the tumor suppressor gene PTEN by activating the PI3K signaling pathway, thereby upregulating the transcriptional activity of RUNX2 in VSMCs (*Liu et al., 2017*). MicroRNA-30b and microRNA-30c have been shown to downregulate the expression of RUNX2 in VSMCs (*Balderman et al., 2012*). *Jeong et al. (2019)* found that the overexpression of the long non-coding RNA Lrrc75a-as1 can downregulate the mRNA levels of various osteoblast-related factors such as BMP2, RUNX2 and MSX2 during the osteogenic transdifferentiation process in VSMCs.

Apart from the above-mentioned epigenetic modification mechanisms, current research has found that histone modifications also play an important role in the regulation of RUNX2 expression. The discovered histone deacetylases (HDACs) are roughly divided into four classes: I, II, III and IV, among which class II HDACs primarily regulate the expression of RUNX2 (*Eom & Kook, 2014*). *Abend et al. (2017)* found that HDAC4 is upregulated in the early stage of osteogenic transdifferentiation in VSMCs, significantly increasing the expression of the VC marker factors RUNX2 and OPN, while knockdown of HDAC4 by shRNA inhibits the progression of VC. Research by *Malhotra et al. (2019)* found that in human aortic VSMCs induced for VC, the expression of HDAC9 is increased, thereby upregulating the expression of RUNX2, and knocking down HDAC9 expression with siRNA can inhibit RUNX2 expression and thus VC. Interestingly, HDAC5 and HDAC6, which belong to the same class II HDACs, seem to downregulate the expression of RUNX2 during the osteogenic transdifferentiation of VSMCs (*Kwon et al., 2020*).

Therefore, it is of great significance to thoroughly investigate the intracellular signaling pathways that upregulate Runx2 expression and the various PTMs that affect RUNX2 activity. This will provide new perspectives for exploring treatments that can

inhibit the upregulation of Runx2 expression in VSMCs and thereby reverse osteogenic transdifferentiation.

## THE INTERACTION BETWEEN HYPERPHOSPHATEMIA, BMP2, AND RUNX2 IN THE PROGRESSION OF VC

In this review, we have discussed the regulatory roles of hyperphosphatemia, BMP2 and RUNX2 in the progression of vascular calcification. It has been observed that there seems to be a certain interaction among these three in some signaling pathways, and this interaction plays an extremely important role in the regulatory mechanisms of VC progression. Research by *Liu et al. (2021)* and *Zhu et al. (2018)* confirmed that the mutual regulation between BMP2 and RUNX2 is realized through the activation of the BMP2/Smad1/5/RUNX2 signaling pathway. The binding of BMP2-mediated Smad1/5 to the nuclear matrix targeting signal domain of RUNX2 can promote the phosphorylation of RUNX2 (*Liu et al., 2021*; *Zhu et al., 2018*), and the activated Smad1/5 can also induce the acetylation of RUNX2 to inhibit the degradation of ubiquitinated RUNX2 (*Jun et al., 2010*). Therefore, BMP2 influences the stability and transcriptional activity of RUNX2 by mediating the activation of Smad1/5, thus regulating the osteogenic transdifferentiation of VSMCs. Consequently, Smad1/5 can be considered as an intermediary that mediates the interaction between BMP2 and RUNX2.

Current research has found that microRNAs also seem to play an important role in the interconnection between BMP2 and RUNX2. BMP2 promotes osteogenic transdifferentiation of VSMCs by downregulating the expression of microRNA-30b and microRNA-30c, which reduces the inhibitory effect of these microRNAs on the expression of RUNX2 (*Balderman et al., 2012*). Additionally, high concentrations of phosphate can upregulate the expression of BMP2 in VSMCs (*Sage et al., 2011*; *Zhang et al., 2017*), and BMP2, in turn, can enhance the uptake of extracellular Pi by VSMCs through the upregulation of Pit-1 expression on the VSMC membrane, further promoting the pathological accumulation of calcium and phosphate and osteogenic transdifferentiation (*Li, Yang & Giachelli, 2008*). When the extracellular Pi level in VSMCs is elevated, activation of signaling pathways like ERK1/2, NF-kB and WNT/$\beta$-catenin upregulate the expression of RUNX2 (*Li, Yang & Giachelli, 2006*; *Yao et al., 2015*; *Yoshida et al., 2017*), which can then mediate an increase in the expression of osteogenic markers such as BMP2, indirectly promoting the uptake of Pi by VSMCs (*Voelkl et al., 2019*).

In summary, hyperphosphatemia can be seen as the initiating factor in a series of regulatory mechanisms during the VC process secondary to CKD, mediating the expression of BMP2 and RUNX2, which are crucial molecules affecting the VC process. Furthermore, BMP2 and RUNX2 can influence the Pi uptake function of VSMCs through various mechanisms. The individual regulatory roles of these three factors and their complex interactions play an extremely important role in the network of regulatory mechanisms for VC.

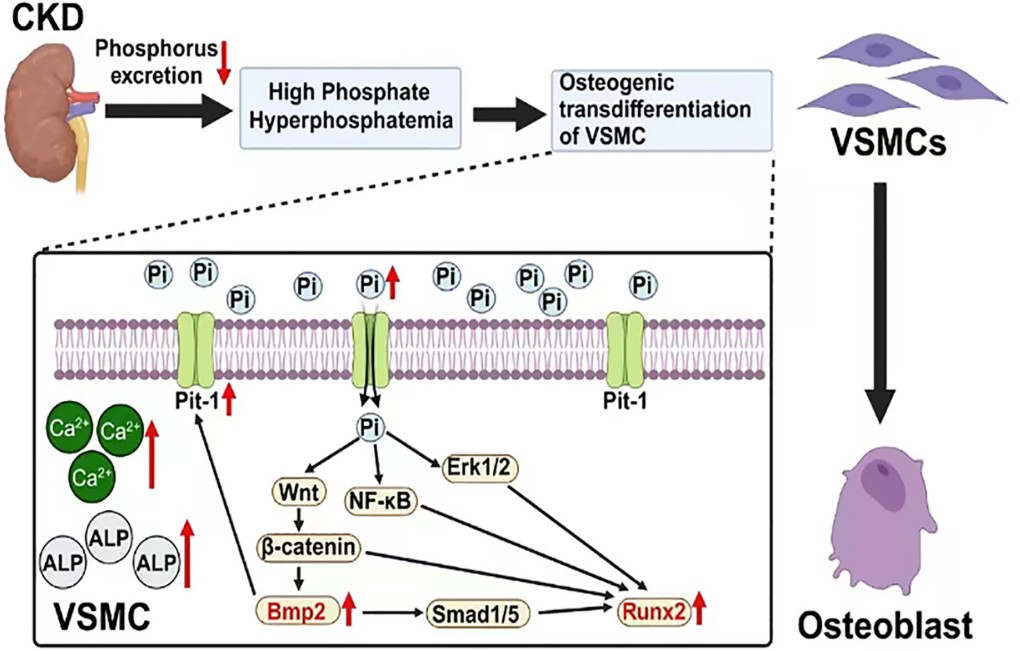

**Figure 1** **The mechanism of vascular calcification involving BMP2 and RUNX2.** Created with Bioren­der.

## CONCLUSION AND OUTLOOK

Vascular calcification is a decisive factor affecting the incidence of cardiovascular events and mortality in the CKD population. During the late stages of CKD, when renal function is severely compromised, serum phosphate levels rise rapidly. The excess extracellular Pi enters the cells *via* sodium-phosphate cotransporters on the VSMC membrane, activating various intracellular Pi-dependent signaling pathways, which initiate the osteogenic transdifferentiation of VSMCs and VC.

BMP2 and RUNX2 serve as key regulatory factors in the process of osteogenic transdifferentiation of VSMCs. They act as downstream molecules with decisive regulatory roles in the series of Pi-dependent signaling pathways that promote VSMC osteogenic transdifferentiation. BMP2 and RUNX2 serve as key factors in the osteogenic transdifferentiation process of VSMCs. Epigenetic modifications occurring at the gene level of these two factors may play a crucial role in the regulatory mechanism. For instance, the acetylation modification of RNA mediated by the recently discovered RNA-binding protein NAT10 has been proven to enhance RNA structural stability, thereby promoting its expression. Based on current research findings, it is speculated that NAT10 may have a regulatory effect on the gene expression of BMP2 and RUNX2 (*Arango et al., 2018*; *Yang et al., 2021*). This offers a new perspective for further studies on the molecular mechanisms behind the osteogenic transdifferentiation of VSMCs. In addition, research has found that losartan (a widely used angiotensin II receptor blocker) may inhibit the progression of VC in rats by downregulating the expression of BMP2, RUNX2, and the angiotensin II

type 1 receptor (AT1R) in VSMCs. This may provide new research directions for further exploring the specific mechanisms of this drug and for developing clinical treatment strategies targeting BMP2 and RUNX2-mediated VC (Li et al., 2016). In conclusion, these factors are part of a regulatory network centered on Pi that mediates the uptake of extracellular Pi by VSMCs. The complex interactions among Pi, BMP2 and RUNX2 indirectly enhance their positive regulatory effects on the osteogenic transdifferentiation of VSMCs and VC (Fig. 1).

Therefore, more research data is needed to clarify whether Pi, BMP2 and RUNX2 have other potential direct effects on VC and to understand their mechanisms of action. This review will provide new insights and evidence for exploring rational and effective new targets for the clinical treatment of VC secondary to CKD. Identifying such targets could lead to the development of novel therapies that may interrupt the pathological processes underpinning VC in CKD patients, thereby reducing associated morbidity and mortality.

### Funding
This work was supported by grants from the National Natural Science Foundation of China (82000651, 82170691, 81974095 and 82370705), the Guangdong Basic and Applied Basic Research Foundation (2022A1515012209), the Research Start-up Funds for High-level Talents in the Affiliated Hospital of Guangdong Medical University (GCC2022038), the Guangdong Provincial Key Laboratory of Autophagy and Major Chronic Noncommunicable Diseases (2022B1212030003), and the National Clinical Key Specialty Construction Project (Institute of Nephrology, Affiliated Hospital of Guangdong Medical University). The funders had no role in study design, data collection and analysis, decision to publish, or preparation of the manuscript.

### Grant Disclosures
The following grant information was disclosed by the authors:
National Natural Science Foundation of China: 82000651, 82170691, 81974095 and 82370705.
Guangdong Basic and Applied Basic Research Foundation: 2022A1515012209.
Research Start-up Funds for High-level Talents in the Affiliated Hospital of Guangdong Medical University: GCC2022038.
Guangdong Provincial Key Laboratory of Autophagy and Major Chronic Noncommunicable Diseases: 2022B1212030003.
National Clinical Key Specialty Construction Project.

### Competing Interests
The authors declare there are no competing interests.

## Author Contributions

- Xinhua Liang conceived and designed the experiments, performed the experiments, analyzed the data, prepared figures and/or tables, authored or reviewed drafts of the article, and approved the final draft.
- Yankun Li analyzed the data, prepared figures and/or tables, and approved the final draft.
- Peng Wang conceived and designed the experiments, authored or reviewed drafts of the article, and approved the final draft.
- Huafeng Liu conceived and designed the experiments, authored or reviewed drafts of the article, and approved the final draft.

## Data Availability

This is a literature review.

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
