# Peer review of "Key regulators of vascular calcification in chronic kidney disease: Hyperphosphatemia, BMP2, and RUNX2"

_PeerJ, doi:10.7717/peerj.18063_

## Round 0.1 · original submission · Major Revisions

For a better understanding of the subject, illustration of the manuscript with at least one figure and some tables is required.

In addition, please carefully check the language and typos.

Reviewer 1 ·

Basic reporting

This paper discusses the mechanisms of VC secondary to CKD, focusing on hyperphosphatemia and its consequent effects on the BMP2 and Runx2 signaling pathways. While it highlights these factors as playing a significant role in vascular calcification, it seems to explain the mechanisms from too narrow a perspective. I think that the explanation of the mechanisms of VC (subtitle 3 and 4) required more detail from a general perspective.

In the section from line 121, "Under normal circumstances," to line 124, "As renal function declines," there is a lack of explanation for why high phosphorus levels occur in patients with end-stage CKD. It merely states that increased phosphate levels lead to increased vascular calcification, without explaining the cause. Therefore, the content appears to lack context.

Experimental design

no comment

Validity of the findings

This paper discusses the role of high phosphate/BMP2/Runx2 in the VC induction process in end-stage CKD patients, but it seems to lack providing potential strategies for subsequent research and exploration of clinical treatments. It merely lists the outcomes that appear phenomenologically, and it is disappointing that the content concludes without additional commentary on BMP2 or Runx2.

Additional comments

It would be beneficial to represent the regulators or inhibitors of BMP2 or Runx2 in diagrams or tables.

Reviewer 2 ·

Basic reporting

the manuscript titled Key regulators of vascular calcification in chronic kidney disease: Hyperphosphatemia, BMP2, and RUNX2 , Xinhua Liang etc. demonstrated the major molecule mechanisms of vascular calcification secondary to chronic kidney disease. This study summarized valuable current insights into the understanding of vascular calcification, but lack of providing exploration directions.

Experimental design

no comment

Validity of the findings

no commen

Additional comments

Generally, this revision is qualified to be published in the PeerJ. And there are some tips for this paper.
1. There loads lots of full spellings and abrreviations.
2. On the basis of presenting the current research status of vascular calcification, the author needs to propose some new research directions.

---

## Round 0.2 · Minor Revisions

Please add comments on the potential practical applicability.

Reviewer 1 ·

Basic reporting

no comment

Experimental design

no comment

Validity of the findings

It would be beneficial to illustrate the mechanisms by which RUNX2 and BMP regulate vascular calcification in a CKD model. Additionally, there is no presentation of clinical therapeutic strategies utilizing this information

---

## Round 0.3 · Minor Revisions

As the reviewer suggested at the second round of revision, please illustrate the involvement of BMP2 and RUNX2 into the mechanisms of vascular calcifications (meaning: please add a schematic Figure).

---

## Round 0.4 · accepted · Accept

The authors have resolved all the previous concerns. No further comments.